# Fatty Acid and Multi-Isotopic Analysis (C, H, N, O) as a Tool to Differentiate and Valorise the Djebel Lamb from the Mountainous Region of Tunisia

**DOI:** 10.3390/molecules28041847

**Published:** 2023-02-15

**Authors:** Samir Smeti, Federica Camin, Luana Bontempo, Souha Tibaoui, Yathreb Yagoubi, Silvia Pianezze, Edi Piasentier, Luca Ziller, Naziha Atti

**Affiliations:** 1Laboratoire de Productions Animales et Fourragères, INRA-Tunisia, University of Carthage, rue Hédi Karray, Ariana 2049, Tunisia; 2International Atomic Energy Agency, Vienna International Centre, P.O. Box 100, A-1400 Vienna, Austria; 3Centro di Ricerca e Innovazione-Fondazione Edmund Mach, via E. Mach, 1, 38098 San Michele all’Adige, TN, Italy; 4C3A-Center Agriculture Food Environment, University of Trento, via E. Mach, 1, 38098 San Michele all’Adige, TN, Italy; 5Centro di Trasferimento Tecnologico-Fondazione Edmund Mach, via E. Mach, 1, 38098 San Michele all’Adige, TN, Italy; 6Dipartimento di Scienze Agroalimentari, Ambientali e Animali, Università Degli Studi di Udine, Via Delle Scienze 206, 33100 Udine, UD, Italy

**Keywords:** fatty acid profile, lamb, isotope ratio mass spectrometry, isotopic ratio, traceability

## Abstract

The objective of this study was to distinguish between the Tunisian Djebel lamb meat and meat from typical Tunisian production systems (PSs) through the fatty acids (FAs) profile and the stable isotope ratio analysis (SIRA). Thirty-five lambs from three different regions and PSs (D = Djebel, B = Bou-Rebiaa, and O = Ouesslatia) were considered for this purpose. The results demonstrated that the PS and the geographic origin strongly influenced the FA profile of lamb meat. It was possible to discriminate between the Djebel lamb meat and the rest of the dataset thanks to the quantification of the conjugated linoleic acids (CLA) and the branched chain FAs. Moreover, statistically different concentrations of saturated, monounsaturated and polyunsaturated FAs and a different n-6/n-3 ratio were found for grazing (D and BR) and indoor (O) lambs, making it possible to discriminate between them. As for the stable isotope ratio analysis, all parameters made it possible to distinguish among the three groups, primarily on the basis of the dietary regimen (δ(^13^C) and δ(^15^N)) and breeding area (δ(^18^O) and δ(^2^H)).

## 1. Introduction

There is a wide range of production systems (PSs) used to produce sheep meat. These depend on grazing quality and availability, or on concentrate source and level, which appropriately determine meat quality [1]. For sheep, as for beef, the muscles of animals consuming grazed pasture contain higher levels of polyunsaturated fatty acids (PUFA) and conjugated linoleic acids (CLA) than those fed on concentrates [1]. These fatty acids (FAs) are considered to be beneficial to human health, which also requires meat that is generally lower in fat, safe, and healthy. Moreover, the PS plays a key role in characterizing an animal’s nutritional background, such as organic or grass-based diets, and can have an impact on the composition and quality of animal products.

Most people believe that only meat from sheep and beef raised on grasslands and natural pastures is authentic and it is often considered to be of superior quality [2,3]. In Tunisia, especially during religious ceremonies, specific lamb types raised on natural pastures such as Sidi-Bouzid or the Djebel mountain-forest lambs are appreciated and sought after by consumers [4,5]. This choice is linked to a unique meat flavour associated with trends or popular memories about natural products of animal origin [6]. Given the societal expectations for this sheep meat, the availability of reliable characterisation methods and suitable authentication procedures are advisable to nurture and valorise the identity of Djebel lamb over other lamb types. This requirement goes beyond the local dimension, since consumers are becoming more interested in the provenance and regional identity of animal products. The reasons for this depend on various factors, including culinary and sensory qualities, purported health benefits associated with regional products, or a decline in consumer confidence over the quality and safety of food produced outside their local region [6].

As a result, the authentication of meat products requires the development of innovative analytical approaches. Few studies focusing on the traceability of grazing lambs by using chemical composition, fatty acid profile, antioxidation stability [7], visible reflectance spectroscopy [8] and stable isotope analysis [9,10,11,12,13] are available in the literature. The last technique offers a time-saving and effective way to verify the authenticity and geographic origin of various products [14] and, thanks to its robustness, it has also been used to build databanks for the traceability of different types of products, such as wine and cheese [14]. In fact, since isotopic features are transferred from feeds to animal products, the isotopic values of animal matrices are expected to correlate with their feeding regimen and geographic origin. Dietary components are defined by signature isotopes determined by climate (H and O), vegetation composition (C, N), feed type (C, N), crop production practices (N, S), and proximity to the sea (S) [11,15]. Therefore, stable isotope ratio analysis is one of the best techniques to detect dietary differences and origin. It was also established that FA profiles for lamb meat varied across geographic regions. This makes it possible to differentiate lamb meat according to different systems or areas using the FA profile [16].

Given the paucity of approved information available on the distinctiveness of Djebel lamb meat, the objective of this study was to use the FA profile and stable isotope ratio to distinguish the uniqueness of Djebel lamb meat reared in mountain-forest regions and on pasture farming systems from the meat of lambs originating from common Tunisian rearing systems, which are based on intermittent pasture supplemented with hay and concentrate. Thirty-five lambs from three different areas of Tunisia were taken into consideration in this study to determine whether stable isotope ratio analysis can corroborate the FAs profile data and ultimately provide additional information to facilitate a geographical distinction.

## 2. Results and Discussion

### 2.1. Fatty Acid Profile

As shown in Table 1, which incorporates the three farming systems (D = Djebel, B = Bou-Rebiaa, and O = Ouesslatia), the predominant saturated fatty acids (SFA) were C16:0 (27.9–29.4%) and C18:0 (16.1–20.7%); on the other hand, C18:1n–9 was the most abundant (25.5–32.3%) among the monounsaturated fatty acids (MUFA). In comparison to the D and BR grazing lambs, the indoor lamb (O) had a higher level of these three FAs. They accounted for 82% vs. 70% of total detected FA for indoor vs. grazing lambs. The prevalence of palmitic, stearic, and oleic acids is consistent with typical values for sheep meat [17,18,19].

Total SFA content was similar across all farming systems, even though lower C16:0 and C18:0 relative amounts have been found for grazing lambs (D and BR) with respect to indoor ones (O), as previously mentioned. The higher value of C14:0 in pasture compared to indoor lambs (7.0% vs. 2.4%) may explain why there is no significant difference among groups as for this FA. The MUFA content was higher (*p* < 0.05) for indoor (O) than for grazing animals (D and BR), due to the higher relative amount of C18:1n-9c of the first group. The total PUFA content was similar for all lambs; however, C18:2n-6 levels were higher for indoor lambs (O) than for grazing lambs (D and BR), while the opposite occurred in the case of C18:3n-3, which resulted in a lower n-6/n-3 ratio for grazing animals, especially the Djebel ones (2.9%). For indoor animals (O), the n-6/n-3 ratio was very high (11.45%), confirming that a diet based on pasture rather than concentrate increase this ratio [20,21]. As an increase in the n-3 PUFA concentration (resulting in a lower n-6/n-3 ratio) is desirable from a human health perspective [19,20], meat produced by grazing lambs such as the Djebel, having the lowest n-6/n-3 of the dataset, should be preferred.

Meat from Djebel lambs had the highest CLA content of the dataset, ten times higher than that of meat from indoor lambs (Table 1). This difference may be explained by the abundance of woody and herbaceous plants rich in polyphenols, which inhibit oxidase activity [22] and limit the total FA bio hydrogenation in the rumen [23], found in the mountain forests where Djebel lambs graze. Our results confirmed the data in the literature on the higher CLA content in the meat of grazing lambs compared to feedlot lambs [20,21,24]. Additionally, Djebel lambs had higher levels of trans-vaccenic acid (C18:1t-11), a precursor for CLA, than any other lamb, confirming the importance of endogenous synthesis of CLA. Our findings are consistent with the literature on the high C18:1t11 and CLA accumulation in meat with diets having a high grass/concentrate ratio [20,21].

Another noticeable distinction between the FA profiles of the farming systems was recorded for the branched chain FAs (BCFA), which had higher values for Djebel than both other systems, especially the indoor one, having less than half the BCFA of Djebel meat. It was shown that augmenting the forage/concentrate ratio in the diet (the case of D and BR in our study) led to a higher proportion of BCFA [25]. Therefore, the BCFA concentration may be used to differentiate the origin of lamb meat in relation to the diet of lambs. This finding supports that of Gómez-Cortés et al. [26] on the use of BCFA as a meat origin discriminant.

The FA profile of lamb meat was strongly influenced by the PS and, therefore, the geographic provenance. In fact, the SFA C20:0, the C18:1t, Cis-9-trans-11-CLA, and BCFA (Table 1) made it possible to distinguish Djebel lamb meat among the three PSs under study (D, BR, and O). Therefore, our results are consistent with recent studies concerning the possibility to identify the dietary background of lambs by means of a discriminant analysis using the FA profile of meat [26,27]. Furthermore, Monahan et al. demonstrated that FA analysis is a successful strategy to differentiate between beef from pasture-raised animals, from those fed a barley-based concentrate, or from those whose diet consisted of grass/concentrate combinations. The model obtained enabled a segregation of the four dietary treatments with a correct classification of 93% [9]. Similarly, Alfaia et al. successfully used the FA profile of meat to identify which animals were fed a barley-based concentrate diet in feedlot, those that were pasture-fed, or those that were pasture-fed and subsequently fed a barley-based concentrate [28].

Moreover, it has also been demonstrated that using the nutritional composition of meat (FA, carotenoids, and vitamin E) associated with the consumption of a particular diet or a specific PS can be used to characterize the dietary background of animals [9]. This is the case for this study, which compares mountain-forest pasture (D) with other feeding systems (O and BR). Alpine forage and leguminous-rich pastures are generally botanically different and result in lambs with higher levels of PUFA in both intramuscular and subcutaneous fat [9]. This is consistent with the results of the current study, where grazing in natural forests dominated by woody species (D lambs) and herbaceous pastures (BR lambs) resulted in meat with a higher PUFAn-3 and CLA concentration. High PUFA levels are known to potentially alter the flavour of meat due to their tendency towards oxidation. Nevertheless, grazing pastures rich in aromatic plants can be of interest since their secondary compounds can exert antioxidant activity of lipids due to their phenolic diterpene content [29].

### 2.2. Stable Isotope Ratio Analysis

The isotopic ranges of values obtained, and the associated statistical results, are reported in Figure 1.

#### 2.2.1. Carbon and Nitrogen Stable Isotopes

It is well established that the main factor affecting the carbon isotope ratio of meat is the proportion of C3 and C4 plants (having different isotopic ranges) in the diet [12]. Indeed, it is known that C4 plants have δ^13^C values between −14 and −12‰, while C3 plants values range between −30 and −23‰ [30]. The D group lambs, which graze mostly in natural forests, had the lowest δ(^13^C) in the dataset (−26.7 ± 0.2‰), as would be expected from a C3-based diet (Figure 1). On the other hand, the O group had the highest δ(^13^C) in the dataset (−20.3 ± 0.3‰) due to the inclusion of a consistent proportion of a C4-based concentrate in the diet. Finally, the BR values (−24.0 ± 0.3‰) fell in between the D and O groups, since the lambs in this region are fed a mixed diet based on equal proportions of both C3 and C4 ingredients. The δ(^13^C) results are consistent with previous studies on lambs’ fed diets that were C3-based [13,31,32], mixed C3-C4 and C4-based [33,34]. In conclusion, the δ(^13^C) made it possible to differentiate between the three groups and figures therefore as a feasible and effective tool to be used as for traceability purposes.

In agreement with other research reported in the literature [33,34], the δ(^15^N) made it possible to differentiate the groups based on their different feeding regimens, figuring as a source of supporting information additional to the δ(^13^C) values (Figure 1). The nitrogen isotope signature in the diet is the main influencing factor in determining the δ(^15^N) of the meat. Indeed, the fertilization practices used to grow the plants [35], the inclusion in the diet of leguminous plants [11] and eventually of products coming from the marine ecosystem [36] can influence the δ(^15^N) of the feed and thus of the meat. In the case of this study, the δ(^15^N) of the plants constituting the pasture (forest for the D group and meadow for the BR group) and the ingredients that make up the concentrate (fed to both the BR and O groups) influence meat δ(^15^N). As the δ(^15^N) values of all groups range from 5.2‰ and 7.7‰, it might be possible to exclude the use of synthetic fertilizers in the area where all lambs were bred, as these products have δ(^15^N) ranging from +4 and −4‰ [35].

It should also be noted that previous research on the isotope ratios of lambs that were allowed to graze in different European countries [11,13] reported significant differences that were not related to the feeding regimen but rather to environmental parameters. In these studies, meat from lamb breeds raised in different countries but fed similar diets had different values of δ(^15^N). Further studies revealed that arid climatic conditions are linked to relatively high values of δ(^15^N) in meat, whereas humidity and a cold climate result in low values [11,12,13,31,37,38]. In accordance with these results, the BR group, which is 46 km from the sea and 80 m above sea level (a.s.l.), has the highest values across the entire dataset, followed by the D group (37 km from the sea but 350 m a.s.l.), and the O group (151 km from the sea and 550 m a.s.l.) (Figure 1).

#### 2.2.2. Oxygen and Hydrogen Stable Isotopes

The hydrogen and oxygen isotope ratios in the meat of animals are influenced by the feed and drinking water. In turn, these parameters are closely linked to factors such as the sampling point’s altitude, latitude, and proximity to the sea [12]. As for carbon and nitrogen, the isotope ratios of hydrogen and oxygen made it possible to discriminate between the three groups (Figure 1).

The BR group’s δ(^2^H) and δ(^18^O) values are the highest in the dataset (−84.9 ± 2.6‰ and 21.0 ± 0.5‰, respectively) (Figure 1). The proximity to the sea (46 km) and the low altitude (80 m a.s.l.) result in warm climatic conditions, leading to relatively high δ(^2^H) and δ(^18^O) values. The BR group values are consistent with previous results obtained for lambs grazing in Southern Italy [11]. Despite being quite close to the coast (37 km), the D site is located 350 m a.s.l., resulting in relatively low δ(^2^H) and δ(^18^O) values (−95.8 ± 2.0‰ and 18.1 ± 0.4‰, respectively) (Figure 1). Data for the D group are similar to previous results on lambs grazing in mountain areas [15,31]. Finally, the δ(^2^H) values of lambs grazing in the O site fell between those of the other groups (−88.4 ± 2.5‰) (Figure 1). On the other hand, the O group’s δ(^18^O) is the lowest in the dataset (−17.3 ± 0.4‰). This is due to the fact that animals that are fed a dry concentrate-based diet are more likely to drink 18O-depleted tap water, resulting in lower values of δ(^18^O) in their meat [39]. In conclusion, the set formed by the four parameters that have been considered in the isotopic analysis (δ(^13^C), δ(^15^N), δ(^18^O) and δ(^2^H)) proved to be effective in providing a method to characterize the Djebel lamb meat among other competitor meat products.

#### 2.2.3. Principal Component Analysis

To optimize the visualization of the results obtained through the SIRA, a Principal Component Analysis (PCA) was performed on the isotopic ratio dataset. The simultaneous representation of the objects and variables projected into the space of PC1 and PC2, which together account for 95.0% of the total variance, is depicted in Figure 2.

Although no information on the meat’s provenance was imported into the PCA, the plot reveals a clear clustering of the three different groups (D, BR, and O). A clear separation of the mentioned groups occurs in a direction almost parallel to PC2 (Figure 2) and both δ^15^N and δ(^18^O), being collinear with it, seem to be the discriminant variables responsible for such separation. At the same time, a separation among groups is also noticeable along PC1 (between D and the rest of the dataset). Finally, the secant of PC1 and PC2 axes collinear to δ(^13^C) seems to isolate the O group from the rest of the dataset, while the secant of PC1 and PC2 axes collinear to δ(^2^H) seems to isolate the BR group. On this basis, the two parameters seem to be the discriminant variables responsible for each separation.

The outcomes confirm, as previously assessed, the effectiveness of isotopic analysis in characterising the Djebel lamb meat by discriminating it against competitor PSs. In general, it has been shown how the SIRA makes it possible to differentiate products from different geographic origins and animals fed different diets, if based on products having distinguishable isotopic fingerprints.

## 3. Materials and Methods

### 3.1. Animals and Rearing Systems

Djebel group lambs (D) were reared in a region characterized by mountainous forest terrain (north-west of Tunisia). Lambs grazed in natural plant forests dominated by woody species and the availability of acorns. Lambs in the region of Bou-Rebiaa (BR) grazed on herbaceous fallow pastures supplemented with hay and concentrate. However, lambs in the region of Oueslatia (O) were reared in a traditional indoor system based on hay and concentrate, with occasional grazing depending on the availability of herbaceous plants. The concentrate mentioned above was composed of barley (55%), maize (30%), soya bean meal (12%), and a blend of vitamins and minerals (3%). The difference between the three experimental sites was based on the grazing site’s distance from the sea and altitude: the rearing site of D group lambs was located in the region of Nefza, 37 km from the sea and at an altitude of 350 m; the second site (BR) is 46 km from the sea and at an altitude of 80 m; and the third site (O) is 151 km from the sea and at an altitude of 550 m.

### 3.2. Measurements and Analysis

#### 3.2.1. Slaughter and Sampling Procedures

At the age of 6 months, lambs (12 from Djebel, 12 from Bou-Rebiaa, and 11 from Oueslatia) were slaughtered at 25.8, 26.8, and 22.6 kg, respectively, in the abattoir of the National Institute of Agronomic Research of Tunisia-INRAT. All the procedures employed in this study (transport and slaughtering) met ethical guidelines and complied with Tunisian legal requirements in accordance with Law No. 2005-95 (18 October 2005).

Before slaughtering, lambs were made to fast for 12 h with free access to water. Animals were weighed just before slaughter. After slaughter, carcasses were chilled at 4 °C for 24 h and then split longitudinally; the *longissimus thoracis* and *lumborum* muscle (LTL) was removed from the left side of each carcass, trimmed of external fat, and frozen at −20 °C for subsequent meat FA and IRMS analyses.

#### 3.2.2. Fatty Acid Analysis

The quantification of the FAs was carried out in the Laboratoire de Productions Animales et Fourragères, University of Carthage (Tunisia). Total intramuscular fat (IMF) was extracted according to the procedure by Folch et al [40]. Nonadecanoic acid (C19:0) was added as an internal standard to 1.5 g of minced LTL muscle and homogenised in 30 mL of chloroform–methanol mixture (2:1 *v*/*v*) using an Ultra-Turrax homogeniser (T 25 basic; Ika-Werke, GmbH & Co., Staufen, Germany). Whatman filter paper (No. 1820-047) was then used to filter the mixture under vacuum. The extract was washed with 8.5 mL of 0.88% KCl (*w*/*v*), mixed vigorously for 1 min, and then left overnight at room temperature. The organic phase was separated, and the solvents were evaporated under vacuum at 40 °C (Univapo ECH System, UniEquip, Munich, Germany).

Fatty acid methyl esters (FAMEs) were prepared using an HCl methanolic solution. Lipid samples were mixed with 2 mL of hexane and 3 mL of HCl methanolic solution in 20-mL glass tubes with Teflon-lined caps. The mixture was heated at 70 °C for 2 h and then cooled to room temperature. After the addition of 5 mL K_2_CO_3_ (6%, *w*/*v*) and Na_2_SO_4_, FAMEs were extracted in 2 mL of hexane. Samples were kept for 30 min prior to centrifugation at 1006× *g* for 10 min at 20 °C; the upper hexane layer was removed, concentrated under N_2_, and then recovered in hexane.

The FAMEs were measured using a Carlo Erba GC (HRGC 5300 Mega series; Rodano, Milan, Italy) fitted with a flame ionization detector (FID) and automatic sampler (model A200S), where 1 mL of sample was injected in split mode (30:1). The GC was equipped with a 60 m SP-2380 fused silica capillary column (0.25 mm i.d., film thickness 0.25mm; Supelco Inc., Bellefonte, PA, USA). The initial oven temperature was 160 °C, subsequently increased to 260 °C at a rate of 5 °C/min, and then held for 5 min. Helium was used as the carrier gas at a flow rate of 1.2 mL/min. FAMEs were identified using original Supelco standard mixture (Supelco 37 component FAME mix) and conjugated linoleic acid (Sigma-Aldrich, Milan, Italy), quantified using C19:0 as an internal standard and expressed as a percentage of the total identified FA.

#### 3.2.3. Isotope Analysis

The isotope analysis was carried out in the laboratories of Centro Trasferimento Tecnologico, Fondazione Edmund Mach (Italy). In order to obtain the defatted fraction to perform the stable isotope analysis, the fat was extracted from the bulk meat by washing each sample with three 10 mL aliquots of ethyl ether:petroleum ether (1:2, *v*/*v*), followed by drying. The procedure was performed according to [41]. The defatted samples were weighed in tin (0.8 mg for ^13^C/^12^C, ^15^N/^14^N) or silver capsules (0.2 mg for ^2^H/^1^H and ^18^O/^16^O) on a microbalance. The comparative-equilibration method was used for the analysis of ^2^H/^1^H. Samples and standards were left at laboratory air moisture for a minimum of 96 h and then placed in a desiccator with P_2_O_5_ under nitrogen atmosphere. Isotope ratios were all measured using an isotope ratio mass spectrometer (Finnigan DELTA XP, Thermo Scientific, Bremen, Germany) after either total combustion (^13^C/^12^C and ^15^N/^14^N) or complete pyrolysis (^2^H/^1^H and ^18^O/^16^O) in an elemental analyser (Finningan DELTA TC/EA, high temperature conversion elemental analyser, Thermo Scientific).

As per IUPAC protocol, isotopic values are expressed as a delta with respect to the international standard V-PDB (Vienna-Pee Dee Belemnite) for *δ*(^13^C), V-SMOW (Vienna-Standard Mean Ocean Water) for δ(^2^H), and δ(^18^O) and Air (atmospheric N_2_) for δ(^15^N), following Equation (1):(1)$$\delta_{ref}{(}^{i}E{/}^{j}E,sample)=\left[\frac{R{(}^{i}E{/}^{j}E,\;sample)}{R{(}^{i}E{/}^{j}E,\;ref)}\right]-1$$
where *ref* is the international measurement standard, *sample* is the analysed sample and *^i^E/^j^E* is the isotope ratio between heavier and lighter isotopes [42]. Delta values are multiplied by 1000 and commonly expressed in units “per mil” (‰) or, according to the International System of Units (SI), as a “milliurey” (mUr) [43].

Isotopic values were calculated against two standards through the creation of a linear equation. Two internal working standards that were each calibrated using international reference materials were used for ^13^C/^12^C and ^15^N/^14^N. As for ^13^C/^12^C, fuel oil NBS-22 (*δ*(^13^C) = −30.03 ± 0.05‰), sucrose IAEA-CH-6 (δ(^13^C) = −10.45 ± 0.03‰) (IAEA-International Atomic Energy Agency, Vienna, Austria), and L-glutamic acid USGS 40 (δ(^13^C) = −26.39 ± 0.04‰) (U.S. Geological Survey, Reston, VA, USA) were used. As for ^15^N/^14^N, L-glutamic acid USGS 40 (δ(^15^N) = −4.52 ± 0.06‰) (U.S. Geological Survey, Reston, VA, USA) and potassium nitrate IAEA-NO3 (δ(^15^N) = +4.7 ± 0.2‰) were used. Keratins CBS (Caribou Hoof Standard δ(^2^H) = −157 ± 2‰ and δ(^18^O) = +3.8 ± 0.1‰) and KHS (Kudu Horn Standard, δ(^2^H) = −35 ± 1‰ and δ(^18^O) = +20.3 ± 0.2‰) from U.S. Geological Survey were used to normalize ^18^O/^16^O and ^2^H/^1^H values.

Each reference material was measured in duplicate at the start and end of each daily group of sample analyses (each sample was also analysed in duplicate). A control material was also included in the analyses of each group of samples to check the measurement’s validity. The accepted maximum standard deviations of repeatability were 0.3‰ for δ(^13^C) and δ(^15^N), 0.5‰ for δ(^18^O), and 4‰ for δ(^2^H).

#### 3.2.4. Statistical Analysis

Univariate statistical analyses were performed by means of the SAS GLM procedure (2002). A one-way ANOVA was used to test the effect of the rearing system on the meat FA profile and stable isotope ratios, applying Tukey’s Test for post hoc analysis. A Principal Component Analysis (PCA) was performed on the isotope ratio dataset to display the data and detect the clustering of different groups.

## 4. Conclusions

The three PSs under consideration were characterised through the quantification of FAs. The FAs analysis made it possible to make a distinction between the C18:1t, Cis-9-trans-11-CLA, and BCFA of the D, BR, and O systems. The assessment of the Djebel lamb meat’s high quality is supported by parameters such as the PUFA n-6/n-3 ratio and the presence of FA groups (e.g., PUFA n-3, CLA), which are desirable from a human health perspective [19,20].

Moreover, all the isotopic analysis parameters δ(^13^C), δ(^15^N), δ(^2^H), and δ(^18^O) made it possible to distinguish between the groups, proving the effectiveness of this technique for classifying food products according to their geographic origin and the diet that the animals are fed. Nevertheless, a database containing a larger number of samples from more sampling points must be developed as a method to formally assess lamb meat authenticity.

## Figures and Tables

**Figure 1 molecules-28-01847-f001:**
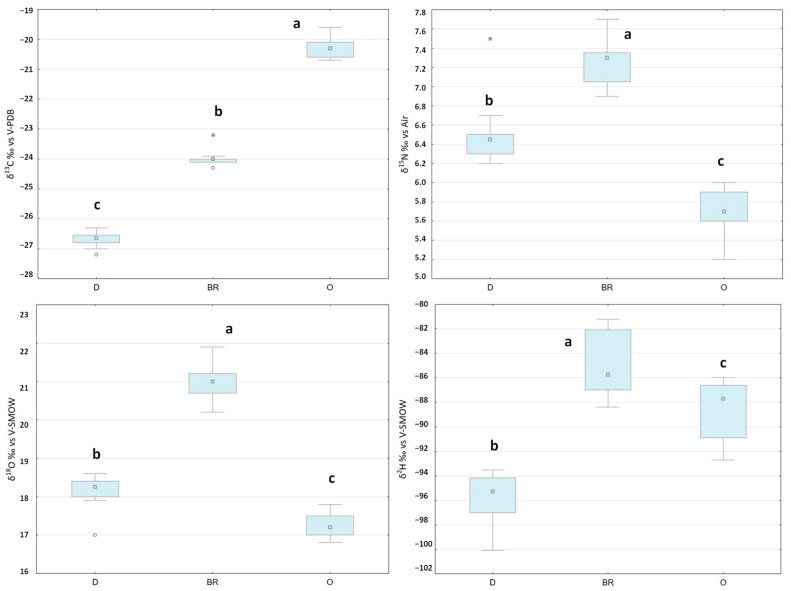
Isotopic value ranges for the three groups: Djebel (D), Bou-Rebiaa (BR), and Oueslatia (O). Statistically different groups (*p* < 0.01), according to the ANOVA, are denoted by different letters (a, b, c).

**Figure 2 molecules-28-01847-f002:**
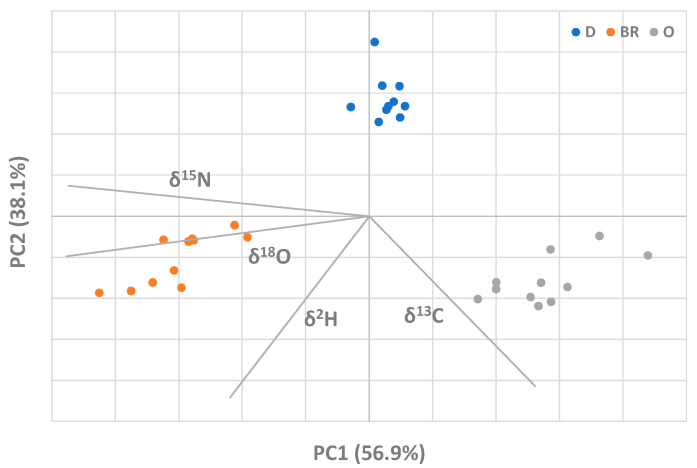
Principal Component Analysis resulting from isotopic results (Djebel (D), Bou-Rebiaa (BR) and Oueslatia (O) groups are labelled by blue, orange and grey spots, respectively).

**Table 1 molecules-28-01847-t001:** The proportion of FAs (percentage of total FAs) in lamb meat based on the farming system. Statistically different groups (*p* < 0.05), according to the ANOVA, are denoted by different letters (A, B, C).

	Djebel	BouRebiaa	Ouesslatia	SEM	*p* Value
C10:0	0.410	0.424	0.289	0.056	0.1976
C12:0	0.976 ^A^	0.926 ^A^	0.113 ^B^	0.087	0.0001
C13:0	0.036 ^A^	0.037 ^A^	0.014 ^B^	0.003	0.0001
C14:0	7.251 ^A^	6.803 ^A^	2.355 ^B^	0.299	0.0001
C15:0	1.411 ^A^	1.247 ^A^	0.376 ^B^	0.060	0.0001
C16:0	27.86 ^B^	28.22 ^AB^	29.44 ^A^	0.447	0.0485
C17:0	2.968 ^A^	2.833 ^A^	1.843 ^B^	0.091	0.0001
C18:0	16.510 ^B^	16.103 ^B^	20.700 ^A^	0.480	0.0001
C20:0	0.084 ^A^	0.066 ^B^	0.039 ^C^	0.005	0.0001
C14:1	0.236 ^A^	0.230 ^A^	0.036 ^B^	0.010	0.0001
C16:1	1.735 ^A^	1.643 ^A^	1.179 ^B^	0.059	0.0001
C17:1	0.587 ^B^	0.697 ^A^	0.505 ^C^	0.021	0.0001
C18:1t	5.023 ^A^	3.390 ^B^	1.512 ^C^	0.239	0.0001
C18:1n-9c	25.45 ^B^	27.52 ^B^	32.33 ^A^	0.718	0.0001
C18:1n-7	0.515 ^C^	0.640 ^B^	0.949 ^A^	0.035	0.0001
C18:1-11t	0.561 ^A^	0.406 ^B^	0.066 ^C^	0.025	0.0001
C18:1-10t	0.205 ^A^	0.131 ^B^	0.047 ^C^	0.014	0.0001
C18:2n-6	2.446 ^B^	2.911 ^AB^	3.740 ^A^	0.303	0.0180
C18:3n-3	1.204 ^A^	0.943 ^A^	0.104 ^B^	0.100	0.0001
Cis-9-trans-11-CLA	1.054 ^A^	0.807 ^B^	0.156 ^C^	0.051	0.0001
C20:2n-6	0.129 ^A^	0.068 ^B^	0.023 ^C^	0.007	0.0001
C20:3n-6	0.034 ^B^	0.062 ^B^	0.183 ^A^	0.014	0.0001
C20:3n-3	0.0272 ^B^	0.041 ^B^	0.081 ^A^	0.006	0.0001
C20:4n-6 ARA	0.364 ^C^	0.718 ^B^	1.812 ^A^	0.121	0.0001
C20:5n-3 EPA	0.151 ^A^	0.191 ^A^	0.049 ^B^	0.030	0.0071
C22:5n-3 DPA	0.229 ^B^	0.371 ^A^	0.299 ^AB^	0.043	0.0773
∑ BCFA	2.753 ^A^	2.470 ^B^	1.199^C^	0.086	0.0001
∑ CLA	1.105 ^A^	0.835 ^B^	0.156 ^C^	0.055	0.0001
∑ SFA	54.92	54.31	53.99	0.682	0.6282
∑ MUFA	34.43 ^B^	34.78 ^B^	36.82 ^A^	0.607	0.0210
∑ PUFA	6.916 ^B^	7.045 ^B^	6.709 ^A^	0.443	0.0004
∑ PUFA n-6	4.200 ^B^	4.664 ^B^	6.021 ^A^	0.396	0.0086
∑ PUFA n-3	1.610 ^A^	1.546 ^A^	0.532 ^B^	0.164	0.0001
n-6/n-3 ratio	2.882 ^B^	3.284 ^B^	11.45 ^A^	0.537	0.0001

## Data Availability

The raw data presented in this study are available on request from the corresponding author.

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
