# Peer review of "Fatty Acid and Multi-Isotopic Analysis (C, H, N, O) as a Tool to Differentiate and Valorise the Djebel Lamb from the Mountainous Region of Tunisia"

_molecules, 2023, doi:10.3390/molecules28041847_

Round 1

Reviewer 1 Report

Abstract - requires a correction (too general). The Abstract does not summarize the paper's discussions.

The premises of the research are very interesting, but I feel unsatisfied after reading the manuscript. The presentation of results and their interpretation are rather weak.  The discussion should be extended. There are reports in the available literature that should be mentioned in the paper. Generally, the section Results and Discussion is limited to the presentation of the observed data, without any significant attempt to explain the causes of the observed regularities.

Author Response

Reviewer 1:

Abstract - requires a correction (too general). The Abstract does not summarize the paper's discussions.

We modified the abstract to better summarise the results of this work, as requested. 

The premises of the research are very interesting, but I feel unsatisfied after reading the manuscript.

The presentation of results and their interpretation are rather weak. 

We modified and improved the Results and discussion part to better describe and comment the results of this study, as requested.

The discussion should be extended.

We modified and improved the Results and discussion part, as requested.

There are reports in the available literature that should be mentioned in the paper.

We added references, as requested.

Generally, the section Results and Discussion is limited to the presentation of the observed data, without any significant attempt to explain the causes of the observed regularities.

We modified and improved the Results and discussion part, as requested.

Reviewer 2 Report

The subject of the manuscript concern traceability in the meat industry with use as a tool for multi-isotopic analysis. The text is clear and well-written. However, there are some general questions to answer to improve the text.

Are other methods of meat identification mentioned?

Whether application of this method is safe and feasible in industrial practice?

Is it economically justified, and what are its advantages over other methods of meat identification?

Some  hints:

Materials and Methods should be before paragraph Results and Discussion

Line 92: table 1 needs a legend explaining the A, B, and C letters.

Line 94-122: hard to read the text, which is not divided into smaller parts. The same problem concerns the following elements of the text.

Line 156: any explanation of letters: a,b, and c in Fig.1.

Line 225: please, add a percentage of concentrate ingredients

Lines 246-269: Please, make subparagraphs, and divide the text into smaller parts.

Line 275:  before you wrote the procedure by Folch et al. [12] and here you wrote, The procedure was performed according to [13]. Something missing?

Author Response

Reviewer 2:

The subject of the manuscript concern traceability in the meat industry with use as a tool for multi-isotopic analysis. The text is clear and well-written. However, there are some general questions to answer to improve the text.

Are other methods of meat identification mentioned?

We would like to clarify that the aim of this work is not to identify a specific meat type (in the present case, lamb), but to provide a method that might be used, after building a databank, to discriminate between Djebel and other competitor lambs.

We improved the Introduction of the manuscript to better explain the aim of this work and to provide other methods used for lamb traceability.

Whether application of this method is safe and feasible in industrial practice?

Hoping to have well understood the question, we would say that the stable isotope ratio analysis (SIRA) and the quantification of the fatty acids could be definitely applied in the industrial practices. In particular, the SIRA represents a robust and effective method for the traceability of animal products, which could be applied both at the real beginning (e.g., in the abattoir) and at the end (e.g., on the final product) of the meat productive chain, in order to trace the product. It could be also used to track the same products, by following each of the production steps from the farm to the fork.

Is it economically justified, and what are its advantages over other methods of meat identification?

As consumers may appreciate and therefore choose animal products deriving from a specific farming system and production factors [1], such in the case of Djebel lamb meat, there absolutely is an economical implication in the possibility to characterise Djebel lamb meat.

We improved the text by adding some information about other methods used for lamb traceability and the advantages of isotope analysis.

[1] (Sañudo et al., 2007; doi:10.1016/j.meatsci.2006.09.009)

Some  hints:

Materials and Methods should be before paragraph Results and Discussion

We ordered the paragraphs according to Molecules authors guidelines.

Line 92: table 1 needs a legend explaining the A, B, and C letters.

We added the information as requested.

Line 94-122: hard to read the text, which is not divided into smaller parts. The same problem concerns the following elements of the text.

We divided in paragraphs the text of line 96-151, as requested.

Line 156: any explanation of letters: a,b, and c in Fig.1.

We added the information as requested.

Line 225: please, add a percentage of concentrate ingredients

We added the information concerning the ingredients, as requested.

Lines 246-269: Please, make subparagraphs, and divide the text into smaller parts.

We divided in paragraphs, as requested.

Line 275:  before you wrote the procedure by Folch et al. [12] and here you wrote, The procedure was performed according to [13]. Something missing?

The quantification of the fatty acids and the isotopic analyses were carried out in two different laboratories. So even though the procedures described in references [12] and [13] aim both to separate the fat from the defatted meat, each laboratory chose the procedure which best fitted the purpose of each lab’s analyses.

We added some information in the text to clarify this concept.

Round 2

Reviewer 1 Report

Accept in present form